

# Leveraging aspect phrase embeddings for cross-domain review rating prediction

Aiqi Jiang[1] and Arkaitz Zubiaga[2]

[1] University of Warwick, Coventry, United Kingdom
[2] Queen Mary University of London, London, United Kingdom

Corresponding author
Aiqi Jiang, A.Jiang.2@warwick.ac.uk, aiqi.jiang@yahoo.com

## ABSTRACT

Online review platforms are a popular way for users to post reviews by expressing their opinions towards a product or service, and they are valuable for other users and companies to find out the overall opinions of customers. These reviews tend to be accompanied by a rating, where the star rating has become the most common approach for users to give their feedback in a quantitative way, generally as a Likert scale of 1–5 stars. In other social media platforms like Facebook or Twitter, an automated review rating prediction system can be useful to determine the rating that a user would have given to the product or service. Existing work on review rating prediction focuses on specific domains, such as restaurants or hotels. This, however, ignores the fact that some review domains which are less frequently rated, such as dentists, lack sufficient data to build a reliable prediction model. In this paper, we experiment on 12 datasets pertaining to 12 different review domains of varying level of popularity to assess the performance of predictions across different domains. We introduce a model that leverages aspect phrase embeddings extracted from the reviews, which enables the development of both in-domain and cross-domain review rating prediction systems. Our experiments show that both of our review rating prediction systems outperform all other baselines. The cross-domain review rating prediction system is particularly significant for the least popular review domains, where leveraging training data from other domains leads to remarkable improvements in performance. The in-domain review rating prediction system is instead more suitable for popular review domains, provided that a model built from training data pertaining to the target domain is more suitable when this data is abundant.

## INTRODUCTION

In recent years, the advent and the prevalent popularisation of social media has led to a change in users' habits of surfing the Internet (*Kaplan & Haenlein, 2010*; *Quan-Haase & Young, 2010*; *Perrin, 2017*; *Goss, 2017*). Since the emergence of social media platforms, Internet users are no longer limited to browsing online contents as mere readers, but they also can also contribute by expressing and sharing their opinions (*Salehan, Zhang & Aghakhani, 2017*). Users can freely post comments and share experiences on target entities

such as products, businesses or events on online review platforms like http://Yelp.com or http://Amazon.com (*Salehan & Kim, 2016*; *Xing et al., 2019*). These reviews present the subjective opinions of people on products or businesses, which are invaluable for consumers and companies (*Sparks & Browning, 2011*). Given the volume of these reviews and the fact that they are spread on different sites across the Internet, it becomes more challenging and costly to aggregate all the reviews on a particular product or business (*Zhang & Qu, 2013*). To alleviate the cost of this task, there is a need to explore the development of automated review rating prediction systems.

There has been work in the scientific literature looking at review rating prediction (*Li et al., 2011*; *Fan & Khademi, 2014*; *Seo et al., 2017a*; *Wang et al., 2018*; *Xing et al., 2019*; *Wang et al., 2019*). This work has however been limited to the prediction of ratings of reviews in very popular domains, such as restaurants (*Ganu, Elhadad & Marian, 2009*; *Zhang et al., 2018*; *Laddha & Mukherjee, 2018*; *Xing et al., 2019*), hotels (*Zhao, Qian & Xie, 2016*; *Laddha & Mukherjee, 2018*) or movies (*Ning et al., in press*). For these domains, it is relatively easy to get a large-scale dataset from online sources for training a review rating prediction system, thanks to sites like TripAdvisor or Yelp where large collections of rated reviews are publicly accessible. The task can however become more challenging for less popular domains, where the dearth of rated reviews available online inevitably means that the scarcity of labelled data available to train a rating prediction model will be rather limited. Moreover, the variance in vocabulary across different domains makes it more difficult to develop a prediction system that generalises to different domains; while reviewers are expected to use phrases like *well written* or *entertaining book* to express that they liked a book, a different vocabulary is expected to indicate that they liked a dentist, such as *careful treatment* or *clean office*. Our work builds on the idea that these phrases, associated with different aspects that vary across domains, can be effectively leveraged for the review rating prediction if the barriers between domains are reduced.

To the best of our knowledge, review rating prediction for non-popular domains has not been studied in previous work, and our work is the first attempt to do so. We propose to pursue review rating prediction for non-popular domains by developing a cross-domain rating prediction system, where rated reviews from popular domains can be leveraged to build a model which can then be generalised to non-popular domains. To facilitate and ensure the effectiveness of building a model that will generalise to other domains, we propose an approach for generating aspect phrase embeddings and polarised aspect phrase embeddings, where phrases that vary across domains can be brought to a common semantic space by using word embeddings.

In this work we make the following contributions:

- We introduce the first cross-domain review rating prediction system that creates semantic representations of aspect phrases using word embeddings to enable training from large-scale, popular domains, to then be applied to less popular domains where labelled data is limited.
- We perform experiments with 12 datasets pertaining to 12 different types of businesses of different levels of popularity.

- Our analysis shows that, while an in-domain review rating prediction system is better for popular domains, for the least popular domains our cross-domain review rating prediction system leads to improved performance when we make use of aspect phrase embeddings.
- Different from an in-domain prediction system, our cross-domain system can be effectively applied to a wide range of product domains found on the Internet that do not necessarily have proper review rating systems to collect labelled data from.
- Our classifier outperforms the results of A$^3$NCF (*Cheng et al., 2018*), a state-of-the-art rating prediction system, in 11 out of 12 domains, showing its generalisability across domains of different levels of popularity.

## RELATED WORK

There has been a body of research in sentiment analysis in recent years (*Liu & Zhang, 2012*). This research has worked in different directions by looking into lexicon-based approaches (*Hu & Liu, 2004*; *Taboada et al., 2011*), machine learning methods (*Pang, Lee & Vaithyanathan, 2002*; *Ye, Zhang & Law, 2009*; *Tripathy, Agrawal & Rath, 2016*) and deep learning techniques (*Wang et al., 2016*; *Poria et al., 2017*; *Zhang, Wang & Liu, 2018*; *Xing et al., 2019*). Different from the sentiment analysis task, the review rating prediction task consists of determining the score—as a rating between 1 and 5—that a user would give to a product or business, having the review text as input. While this may at first seem like a fine-grained sentiment analysis task, which is a translation of a textual view to a numerical perspective and consists of choosing one of five labels rather than three labels (positive, neutral, negative), there are two key characteristics that make the review rating prediction task different. First, the sentiment of a review is not necessarily the same as the rating, as a user may express a positive attitude when sharing a low rating opinion of a business, e.g., "I very much enjoyed my friend's birthday celebration, however the food here is below standard and we won't be coming back". And second, a review tends to discuss different aspects of a business, some of which may be more important than others towards the rating score, e.g., in a review saying that "the food was excellent and we loved the service, although that makes the place a bit pricey", a user may end up giving it a relatively high score given that key aspects such as the food and the service were considered very positive. In addition, the review can discuss different aspects, and the set of aspects discussed in each review can vary, with some users not caring about certain aspects; e.g., one user focuses on food while another one focuses more on price (*Ganu, Elhadad & Marian, 2009*). Using the same star rating mode to express the score of specific features can be more helpful to find users' interests. Hence, we argue that a review rating prediction system needs to consider the opinions towards different aspects of a business. We achieve this by extracting aspect phrases mentioned in different sentences or segments of a review, which are then aggregated to determine the overall review rating. Despite the increasing volume of research in review rating prediction, which we discuss in what follows, research in cross-domain review rating prediction is still unstudied. This is particularly important for less popular review domains where labelled data is scarce and hence one may need to exploit labelled data from a different, more popular domain for training a classifier.

Review rating prediction has commonly been regarded as a classification task, but also as a regression task on a few occasions (*Li et al., 2011*; *Pang & Lee, 2005*; *Mousavizadeh, Koohikamali & Salehan, 2015*). The text of the review is the input that is consistently used by most existing works (*Qu, Ifrim & Weikum, 2010*; *Leung, Chan & Chung, 2006*; *Mousavizadeh, Koohikamali & Salehan, 2015*), however many of them also incorporate other features extracted from the product being reviewed or from the user writing the review (*Qu, Ifrim & Weikum, 2010*). *Ganu, Elhadad & Marian (2009)* improved the review rating prediction accuracy by implementing new ad-hoc and regression-based recommendation measures, where the aspect of user reviews is considered; their study was however limited to only restaurant reviews and other domains were not studied. A novel kind of bag-of-opinions representation (*Qu, Ifrim & Weikum, 2010*) was proposed and each opinion consists of three parts, namely a root term, a set of modifier terms from the same sentence, and one or more negation terms. This approach shows a better performance than prior state-of-the-art techniques for review rating prediction. Datasets including three review domains were used for their experiments, however they ran separate experiments for each domain and did not consider the case of domains lacking training data. Similarly, *Wang, Lu & Zhai (2010)* performed experiments predicting the ratings of hotel reviews, using a regression model that looked at different aspects discussed in a review, with the intuition that different aspects would have different levels of importance towards determining the rating.

Recent years have seen an active interest in researching approaches to review rating prediction but are still limited to popular domains and do not consider the case of domains lacking training data. In a number of cases, features extracted from products and users are being used (*Jin et al., 2016*; *Lei, Qian & Zhao, 2016*; *Seo et al., 2017b*; *Wang et al., 2018*), which limits the ability to apply a prediction system to new domains and to unseen products and users. *Tang et al. (2015)* studied different ways of combining features extracted from the review text and the user posting the review. They introduced the User-Word Composition Vector Model (UWCVM), which considers the user posting a review to determine the specific use that a user makes of each word. While this is a clever approach to consider differences across users, it also requires observing reviews posted by each user beforehand, and cannot easily generalise to new, unseen users, as well as to new review domains where we lack any information about those users. An approach that predicts review ratings from the review text alone is that by *Fan & Khademi (2014)*. They performed a study of two kinds of features: bag-of-words representations of reviews, as well as part-of-speech tags of the words in the review. They studied how the top K keywords in a dataset contributed to the performance of the rating prediction system, finding that a multinomial logistic regression classifier using the top 100 keywords performed best. Others have used topic modelling techniques like LDA or PLSA to identify the aspects that users discuss in restaurant reviews; however, they did not study their effectiveness for review rating prediction (*Titov & McDonald, 2008*; *Huang, Rogers & Joo, 2014*; *Zhou, Wan & Xiao, 2015*; *Vargas & Pardo, 2018*).

One of the most recent methods by *Cheng et al. (2018)* combines topic modelling with a neural network classifier. The approach, called A$^3$NCF, extracts users' preferences and

items' characteristics on different aspects from reviews. An LDA topic modelling layer is used to then generate topic-specific representations of users and items. Subsequently, the method uses an attention network which combines these user and item features extracted from reviews. This method is shown to outperform an earlier method by *Catherine & Cohen (2017)*, called TransNet, which uses two convolutional neural networks to learn latent representations of users and items. We will use the approach by *Cheng et al. (2018)*, namely $A^3NCF$, as a state-of-the-art baseline classifier.

In this paper, we propose to perform a textual analysis of reviews that goes beyond the sole exploration of the whole text as a unit. Our method also looks at the aspect phrases mentioned in the text, which contain the core opinionated comments. This is the case of *good food* for a restaurant or *comfortable bed* for a hotel. While these differ substantially across review domains, we propose to leverage aspect phrase embeddings to achieve representations of aspect phrases to enable generalisation. This allows us to perform cross-domain review rating experiments, where one can build a rating prediction model out of larger training data thanks to leveraging labelled data from other domains. To the best of our knowledge, ours is the first work to perform rating predictions for review domains with scarce training data, such as dentists, as well as the first to propose and test a cross-domain review rating prediction system.

## DATASETS

To enable our analysis of review rating prediction over different domains, we make use of 12 different datasets, each associated with a different domain. We use three different data sources to generate our datasets: (1) a publicly available collection of 6 million reviews from Yelp (https://www.yelp.com/dataset), (2) a collection of more than 142 million reviews from Amazon provided by *McAuley et al. (2015)*; *He & McAuley (2016)*, and (3) a collection of more than 24 million reviews retrieved from businesses listed in TripAdvisor's top 500 cities. We filter 12 categories of reviews from these 3 datasets, which gives us 12 different datasets that enable us to experiment with review rating prediction over 12 different types of businesses and products. All of the datasets include both the review text and the review rating in a scale from 1 to 5. The use of a standardised 1–5 rating scale facilitates experimentation of rating prediction systems across domains.

Table 1 shows the list of 12 datasets we use for our experimentation. Our datasets comprise more than 58 million reviews overall, distributed across different types of businesses, where some businesses are far more popular than others. The number of reviews per type of business ranges from 22 million reviews for books to 36,000 reviews for dentists, showing a significant imbalance in the size and popularity of different domains. The variability of dataset sizes and popularity of review domains enables our analysis looking at the effect of the availability of large amounts of in-domain data for training.

In Fig. 1 we show a breakdown of the 1–5 star ratings for each dataset. We observe an overall tendency of assigning high ratings across most categories, except in the cases of casinos and resorts, where the ratings are more evenly distributed. Most categories show an upwards tendency with higher number of reviews for higher ratings, as is the case with

Table 1 **Details of the 12 datasets, sorted by number of reviews.** The number of reviews that each type of business/product receives varies drastically.

| Business/product | Source | # Reviews |
|---|---|---|
| Books | Amazon | 22,507,155 |
| Restaurants | TripAdvisor | 14,542,460 |
| Attractions | TripAdvisor | 6,358,253 |
| Clothing | Amazon | 5,748,920 |
| Homeware | Amazon | 4,253,926 |
| Hotels | TripAdvisor | 3,598,292 |
| Nightlife | Yelp | 877,352 |
| Events | Yelp | 387,087 |
| Casinos | Yelp | 115,703 |
| Hair salons | Yelp | 99,600 |
| Resorts | Yelp | 57,678 |
| Dentists | Yelp | 36,600 |
| TOTAL | – | 58,583,026 |

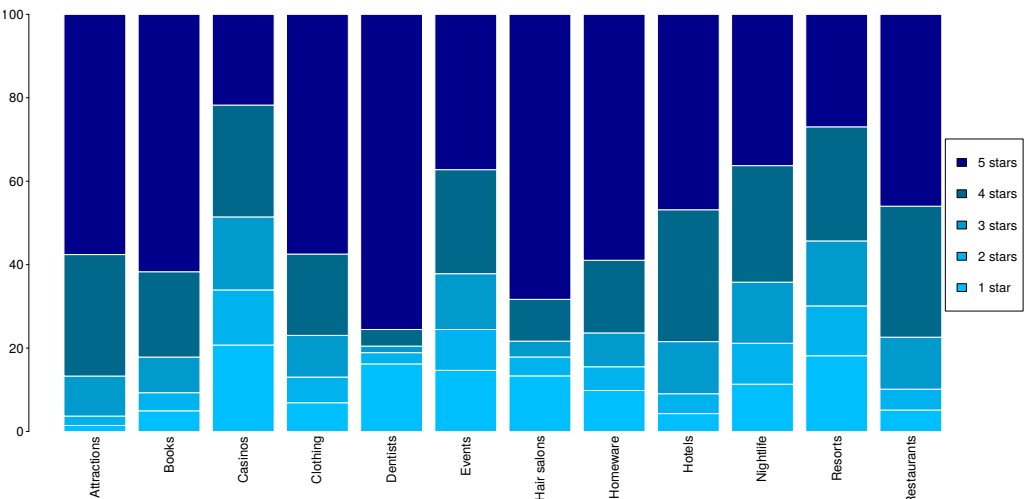

**Figure 1** **Distributions of star ratings in the 12 datasets.** .

attractions, books or restaurants. Interestingly, in the case of dentists and hair salons, the ratings that prevail are 1′s and 5′s, showing that users tend to either love or hate these services.

## METHODOLOGY

One of the key challenges of dealing with reviews pertaining to different domains is that the vocabulary can vary significantly. We can expect that people will express that they like or dislike a product in different ways for different domains. For example, one may make a reference to *good food* when reviewing a restaurant, a *comfortable bed* for a hotel, an *inspiring novel* for a book and a *fun party* for an event. All of these could be deemed

similarly positive for each domain, however without a proper method to capture these semantics, predictions made across domains may not be accurate enough. To tackle this problem, we propose to use aspect phrase embeddings.

## Aspect phrase extraction and representation

Different review domains tend to have different aspect categories associated with them. While one may care about *food quality* in restaurants, the focus is instead on the *engagement* when reviewing a book. Even if other aspect categories such as *price* are more widely generalisable across domains, most aspect categories and associated vocabulary vary across domains. In the first instance, we are interested in capturing those phrases associated with aspect categories, which we refer to aspect phrases. We define aspect phrases as tuples providing opinionated judgement of a particular aspect of a product, e.g., *excellent food* for restaurants or *interesting reading* for books. Once we have these tuples, in a second step we use word embeddings to achieve a generalisable semantic representation of the aspect phrases.

### Aspect phrase extraction

To extract the tuples corresponding to aspect phrases from the reviews, we rely on the assumption that these opinionated tuples will be made of (1) a sentiment word that judges the object or action concerned and (2) a non-sentiment word that refers to the object or action being judged. To restrict the context in which a tuple can be observed, we consider segments of the reviews, i.e., sentences or parts of sentences that are independent from each other. We perform the extraction of aspect phrases by following the next steps:

1. **POS tagging:** We extract the part-of-speech (POS) tags of all words in the reviews by using NLTK's POS tagger (*Bird & Loper, 2004*), hence labelling each word as a noun, verb, adjective, adverb, etc.

2. **Identification of sentiment words:** We use the sentiment lexicon generated by *Hu & Liu (2004)*, which provides a list of over 6,800 words associated with positive or negative sentiment. With this list, we tag matching keywords from reviews as being positive or negative.

3. **Segmentation of reviews:** We break down the reviews into segments. To identify the boundaries of segments, we rely on punctuation signs (, . ; :) and coordinating conjunctions (for, and, nor, but, or, yet, so) as tokens indicating the end of a segment. Text at either side of these tokens are separated into different segments.

4. **Extraction of aspect phrases:** At this stage, we only act within the boundaries of each segment. Within each segment, we identify pairs of words made of (1) a sentiment word labelled as positive or negative and (2) a word labelled as noun or verb by the POS tagger and identified as a non-sentiment word, i.e., not matching any of the keywords in the sentiment lexicon. Each pair of words matching these criteria within a segment is extracted as a tuple pertaining to an aspect phrase.

Through the process above, we extract aspect phrases for all review domains. Table 2 shows the most salient aspect phrases for each of the review domains. We observe that aspect phrases vary substantially across domains, with phrases referring to the *ease of use*

**Table 2   List of most salient aspect phrases for the 12 review categories.**

| Category | Top aspect phrases |
|---|---|
| Books | (great, book) (good, book) (like, book) (recommend, book) (well, written) |
| Restaurants | (good, food) (good, was) (delicious, was) (great, food) (good, service) |
| Attractions | (worth, visit) (beautiful, is) (worth, is) (free, is) (attraction, is) |
| Clothing | (comfortable, are) (well, made) (worn, have) (perfectly, fit) (love, shoes) |
| Homeware | (easy, clean) (easy, use) (great, works) (well, works) (stainless, steel) |
| Hotels | (nice, hotel) (good, hotel) (great, hotel) (recommend, hotel) (clean, was) |
| Nightlife | (happy, hour) (good, was) (pretty, was) (good, were) (good, food) |
| Events | (nice, was) (clean, was) (pretty, was) (clean, room) (nice, room) |
| Casinos | (nice, was) (nice, room) (like, casino) (like, room) (clean, room) |
| Hair salons | (great, hair) (like, hair) (amazing, hair) (best, hair) (recommend, salon) |
| Resorts | (grand, mgm) (lazy, river) (nice, was) (nice, room) (like, room) |
| Dentists | (best, dentist) (wisdom, teeth) (work, done) (clean, office) (recommend, office) |

for homeware, *comfort* for clothing, *food quality* for restaurants or *happy hour* for nightlife, among others.

### Aspect phrase representation

Despite the ability of our method above to capture aspect phrases independent of the domain, these still vary in terms of vocabulary. To achieve semantic representations of aspect phrases extracted for different domains, we make use of Word2Vec word embeddings (*Mikolov et al., 2013*). We train a word embedding model using the 58 million reviews in our datasets. This model is then used to achieve semantic representations of the aspect phrases with the following two variants:

- **Aspect Phrase Embeddings (APE):** we aggregate all the aspect phrases extracted for a review. We generate the embedding vector for each review by adding up the word embeddings for all words included in those phrases.
- **Polarised Aspect Phrase Embeddings (PAPE):** we aggregate the aspect phrases for a review in two separate groups, one containing positive phrases and the other containing negative phrases. Following the same method as for aspect phrase embeddings, here we generate a separate embedding representation for each group, which leads to an embedding representation for positive aspect phrases and another embedding

representation for negative aspect phrases. We then concatenate both embeddings to get the combined vector.

## Experiment settings
### Cross-validation

We perform different sets of experiments for comparing the performance of our rating prediction systems in in-domain and cross-domain settings for different domains. As we are interested in performing 10-fold cross-validation experiments for both settings, we randomly split the data available for each domain $d \in \{1..12\}$ into 10 equally sized folds, $f \in \{1..10\}$. Each time one of these folds is considered as the test data, $Test_{ij}$, while the training data depends on the setting, i.e., (1) $\sum_{\forall f \in \{1..10\}, f \neq j, d = i} Train_{df}$ for in-domain experiments, and (2) $\sum_{\forall d \in \{1..12\}, d \neq i, f = j} Train_{df}$ for cross-domain experiments. We ultimately average the performance scores across all 10 folds in each setting.

### Classifiers

In setting up the experiments for our analysis, we tested all the classifiers proposed by *Fan & Khademi (2014)* and some more, including a multinomial Logistic Regression, a Gaussian Naive Bayes classifier, Support Vector Machines and Random Forests. Our experiments showed that the multinomial Logistic Regression was clearly and consistently the best classifier, and hence for the sake of clarity and brevity we report results using this classifier in the paper.

### Features and baselines

We implement four different sets of features, which include two baselines and two methods that we propose:

- **Baseline (A$^3$NCF):** Introduced by *Cheng et al. (2018)*, A$^3$NCF is a model that extracts users' preferences and items' characteristics on different aspects from reviews. They introduce an attention network which utilises these user and item features extracted from reviews, which aims to capture the attention that a user pays to each aspect of an item in a review. In the absence of cross-domain rating prediction systems in previous work, we use A$^3$NCF as the state-of-the-art rating prediction system to beat.
- **Baseline using Word Embeddings (w2v):** we implement a second baseline consisting of a word embedding representation of the entire text of a review by averaging all the embeddings. We use the word embedding model we trained from our review datasets, which serves as a comparable baseline to measure the impact of aspect phrase embeddings on the performance of the rating prediction system.
- **Aspect Phrase Embeddings (APE):** we concatenate the word embedding vectors obtained for the entire texts of reviews with the aspect phrase embeddings, i.e., word embedding representations of the aspect phrases extracted from a review.
- **Polarised Aspect Phrase Embeddings (PAPE):** we concatenate the word embedding vectors obtained for the entire texts of reviews with the polarised aspect phrase embeddings, i.e., a concatenation of the word embedding representations of positive aspect phrases and the word embedding representation of negative aspect phrases.

Note that baseline methods from more recent works are not directly applicable to our task as they make use of user data, which is not applicable in cross-domain scenarios where users are different across domains and platforms. Because our objective is to build a review rating prediction system that would then be applicable to other sources on the Web, relying on user metadata is not realistic, given that we often do not know the author of a text on the Web or the authors are different from those in the training data. Hence, we use the state-of-the-art text-only review rating prediction system by *Fan & Khademi (2014)*, as well as a baseline using word embeddings, which enables direct comparison with the use of aspect phrase embeddings as additional features.

## Evaluation

The review rating prediction task can be considered as a rating task with five different stars from 1 to 5. As a rating task, we need to consider the proximity between the predicted and reference ratings. For instance, if the true star rating of a review is 4 stars, then the predicted rating of 3 stars will be better than a predicted rating of 2 stars. To account for the difference or error rate between the predicted and reference ratings, we rely on the two metrics widely used in previous rating prediction work (*Li et al., 2011*): the Root Mean Square Error (RMSE) (see Eq. (1)) and Mean Absolute Error (MAE) (see Eq. (2)).

$$\text{RMSE} = \sqrt{\frac{\sum_{i=1}^{n}(\hat{y}_i - r_i)^2}{n}} \tag{1}$$

$$\text{MAE} = \frac{\sum_{i=1}^{n}|y_i - r_i|}{n} \tag{2}$$

where:

$n$ denotes the total number of reviews in the test set.

$y_i$ is the predicted star rating for the $i$th review.

$r_i$ is the real star rating given to the $i$th review by the user.

We report both metrics for all our experiments, to facilitate the comparison for future work.

## RESULTS

The results are organised in two parts. First, we present and analyse results for in-domain review rating prediction, where the training and test data belong to the same domain. Then, we show and analyse the results for cross-domain review rating prediction, where data from domains that differ from the test set is used for training. A comparison of the performance on both settings enables us to assess the suitability of leveraging a cross-domain classifier as well as the use of aspect phrase embeddings.

### In-domain review rating prediction

Tables 3 and 4 show results for in-domain review rating prediction, where the training and test data belong to the same domain. Experiments are performed using 10-fold cross validation within each of the 12 datasets. Note that lower scores indicate a smaller amount

**Table 3** **MAE results for in-domain review rating prediction.** Categories are sorted in descending order by number of reviews, with the most popular review categories at the top of the list.

|  | | | MAE | | |
| | A $^3$NCF | w2v | ape | pape | w2v + ape | w2v + pape |
|---|---|---|---|---|---|---|
| Books | 0.821 | 0.542 | 0.664 | 0.663 | 0.530 | **0.521** |
| Restaurants | 0.779 | 0.484 | 0.598 | 0.611 | 0.471 | **0.466** |
| Attractions | 0.644 | 0.567 | 0.771 | 0.803 | 0.554 | **0.549** |
| Clothing | 0.988 | 0.544 | 0.760 | 0.742 | 0.529 | **0.518** |
| Homeware | 1.055 | 0.581 | 0.760 | 0.751 | 0.563 | **0.543** |
| Hotels | 0.746 | 0.455 | 0.553 | 0.579 | 0.442 | **0.441** |
| Nightlife | 1.061 | 0.579 | 0.691 | 0.715 | 0.555 | **0.554** |
| Events | 1.127 | 0.595 | 0.678 | 0.709 | **0.565** | 0.568 |
| Casinos | 1.127 | 0.769 | 0.771 | 0.803 | **0.689** | 0.705 |
| Hair salons | 1.096 | 0.410 | 0.498 | 0.521 | **0.405** | 0.414 |
| Resorts | 1.208 | 0.793 | 0.760 | 0.777 | **0.689** | 0.692 |
| Dentists | 1.767 | 0.322 | 0.404 | 0.399 | **0.317** | 0.324 |

of error and hence better performance. These results show that both APE and PAPE combined with word embeddings (w2v+APE and w2v+PAPE) consistently outperform the sole use of word embeddings (w2v). We also observe that the sole use of phrase embeddings (APE and PAPE) does not suffice to outperform word embeddings, whereas their combination with w2v leads to clear improvements in all cases. This proves the importance of phrases, however we also see that using the rest of the text in the review is useful to boost performance. Likewise, both of our combining methods clearly outperform the baseline method (A$^3$NCF) by *Cheng et al. (2018)*, with the exception of attractions when the performance is measured using RMSE, which is the only case where the baseline performs better; the combining methods leveraging phrase embeddings are clearly better for the rest of the domains. This emphasises the importance of using word embeddings for capturing semantics of aspect phrases, even when the experiments are within the same domain. A comparison between our two proposed methods using phrase embeddings shows that polarised phrase embeddings outperform phrase embeddings for the most popular review categories, whereas the difference is not so clear for less popular categories.

All in all, these results show that the use of either form of phrase embeddings combined with word embeddings leads to improvements in the review rating prediction when the training and test data belong to the same domain. The main goal of our work is however to show their effectiveness on cross-domain review rating prediction, which we discuss in the next section. The in-domain results presented in this section enable us to perform a comparison with the cross-domain results presented next.

## Cross-domain review rating prediction

Tables 5 and 6 show results for cross-domain review rating prediction. While experiments are also performed using 10-fold cross-validation, we train the classifier for a particular domain using data from the other 11 datasets, i.e., simulating the scenario where we do not have any labelled data for the target domain. We also include results for the best

**Table 4** **RMSE results for in-domain review rating prediction.** Categories are sorted in descending order by number of reviews, with the most popular review categories at the top of the list.

| | A³NCF | w2v | ape | RMSE pape | w2v + ape | w2v + pape |
|---|---|---|---|---|---|---|
| Books | 1.068 | 1.057 | 1.223 | 1.225 | 1.038 | **1.023** |
| Restaurants | 0.976 | 0.856 | 1.007 | 1.031 | 0.836 | **0.828** |
| Attractions | **0.823** | 0.996 | 1.132 | 1.126 | 0.975 | 0.968 |
| Clothing | 1.211 | 1.043 | 1.343 | 1.320 | 1.021 | **1.001** |
| Homeware | 1.281 | 1.144 | 1.381 | 1.370 | 1.115 | **1.081** |
| Hotels | 0.926 | 0.809 | 0.942 | 0.990 | 0.786 | **0.784** |
| Nightlife | 1.251 | 1.001 | 1.139 | 1.172 | 0.961 | **0.958** |
| Events | 1.306 | 1.054 | 1.153 | 1.196 | **1.004** | 1.006 |
| Casinos | 1.304 | 1.245 | 1.218 | 1.252 | **1.118** | 1.128 |
| Hair salons | 1.271 | 1.019 | 1.150 | 1.168 | **0.993** | 0.997 |
| Resorts | 1.395 | 1.295 | 1.224 | 1.238 | 1.143 | **1.136** |
| Dentists | 1.891 | 1.016 | 1.157 | 1.121 | 0.994 | **0.983** |

**Table 5** **MAE results for cross-domain review rating prediction.** BID, best in-domain. Categories are sorted in descending order by number of reviews, with the most popular review categories on top of the list.

| | A³NCF | BID | w2v | ape | MAE pape | w2v + ape | w2v + pape |
|---|---|---|---|---|---|---|---|
| books | 0.913 | **0.521** | 0.725 | 0.858 | 0.826 | 0.691 | 0.672 |
| restaurants | 0.866 | **0.466** | 0.514 | 0.624 | 0.657 | 0.502 | 0.501 |
| attractions | 0.736 | **0.549** | 0.613 | 0.702 | 0.757 | 0.608 | 0.596 |
| clothing | 0.995 | **0.518** | 0.712 | 0.886 | 0.834 | 0.684 | 0.656 |
| homeware | 1.061 | **0.543** | 0.821 | 0.993 | 0.964 | 0.801 | 0.776 |
| hotels | 0.826 | **0.441** | 0.494 | 0.588 | 0.654 | 0.483 | 0.490 |
| nightlife | 1.111 | 0.554 | 0.541 | 0.695 | 0.723 | 0.530 | **0.524** |
| events | 1.155 | 0.565 | 0.543 | 0.682 | 0.729 | 0.525 | **0.521** |
| casinos | 1.151 | 0.689 | 0.652 | 0.799 | 0.851 | 0.630 | **0.623** |
| hair salons | 1.169 | 0.405 | 0.353 | 0.494 | 0.569 | **0.337** | 0.341 |
| resorts | 1.106 | 0.689 | 0.633 | 0.762 | 0.819 | 0.606 | **0.602** |
| dentists | 1.258 | 0.317 | 0.272 | 0.451 | 0.543 | **0.266** | 0.268 |

performance for each review category when we train with data from the same domain, which is represented as BID (best in-domain). This enables us to compare whether and the extent to which the use of out-of-domain data for training can help to improve the performance for each review category.

Results show a remarkable difference between popular and non-popular review categories. For the 6 most popular review categories (books, restaurants, attractions, clothing, homeware, hotels), the best performance is obtained by the best in-domain (BID) classifier, which indicates that for review categories with large amounts of training data, it is better to use in-domain data. However, when we look at the bottom 6 review categories (nightlife, events, casinos, hair salons, resorts, dentists), we observe that the cross-domain

**Table 6** **RMSE results for cross-domain review rating prediction.** BID, best in-domain. Categories are sorted in descending order by number of reviews, with the most popular review categories on top of the list.

| | | | RMSE | | | | |
|---|---|---|---|---|---|---|---|
| | A $^3$NCF | BID | w2v | ape | pape | w2v + ape | w2v + pape |
| books | 1.130 | **1.023** | 1.354 | 1.481 | 1.440 | 1.307 | 1.278 |
| restaurants | 1.121 | **0.828** | 0.891 | 1.045 | 1.094 | 0.870 | 0.868 |
| attractions | 0.902 | **0.823** | 1.062 | 1.155 | 1.275 | 1.050 | 1.034 |
| clothing | 1.237 | **1.001** | 1.265 | 1.453 | 1.418 | 1.229 | 1.190 |
| homeware | 1.330 | **1.081** | 1.443 | 1.600 | 1.579 | 1.417 | 1.386 |
| hotels | 1.073 | **0.784** | 0.854 | 0.980 | 1.077 | 0.836 | 0.844 |
| nightlife | 1.359 | 0.958 | 0.937 | 1.133 | 1.177 | 0.921 | **0.912** |
| events | 1.402 | 1.004 | 0.973 | 1.148 | 1.221 | 0.943 | **0.936** |
| casinos | 1.414 | 1.118 | 1.081 | 1.250 | 1.322 | 1.048 | **1.039** |
| hair salons | 1.437 | 0.993 | 0.900 | 1.116 | 1.246 | **0.866** | 0.876 |
| resorts | 1.382 | 1.136 | 1.064 | 1.214 | 1.296 | 1.024 | **1.018** |
| dentists | 1.513 | 0.983 | 0.887 | 1.181 | 1.323 | **0.870** | 0.873 |

classifier leveraging out-of-domain data for training achieves higher performance. While the sole use of word embeddings (w2v) leads to improved performance for the least popular categories, the improvement is even better when we incorporate either phrase embeddings or polarised phrase embeddings. The results are also positive for the w2v+PAPE over the w2v+APE; even if the results are not better in 100% of the cases, w2v+PAPE tends to outperform w2v+APE in most cases, with just a small difference when APE is better, showing that one can rely on PAPE for all cases dealing with non-popular domains. Our combining methods (w2v+APE and w2v+PAPE) also outperform the state-of-the-art baseline A$^3$NCF for all of the non-popular review domains, with some variation for the popular review domains. This again reinforces the potential of our methods combining phrase embeddings for cross-domain review rating prediction applied to review domains with little training data available. These experiments show a clear shift in the performance of in-domain classifiers when the amount of training data decreases, encouraging the use of out-of-domain data for training in those cases.

Likewise, this motivates the use of the cross-domain classifier for new domains where no labelled data is available for training, for instance because reviews are collected from a website where no ratings are given by the user, such as Facebook, Twitter or comments on blogs.

Further to this, Fig. 2 shows the relative improvements of our w2v+PAPE cross-domain classifier over the A$^3$NCF and the BID classifiers, both for the MAE and RMSE metrics. Review domains are sorted by popularity on the $x$ axis, with the least popular domains on the right. The relative improvement on each review domain is represented by the blue line. The line of best fit (in black) shows the overall tendency of our cross-domain classifier to achieve an increasing relative improvement over other methods as the popularity of the review domain and subsequently the amount of in-domain training data increases. This

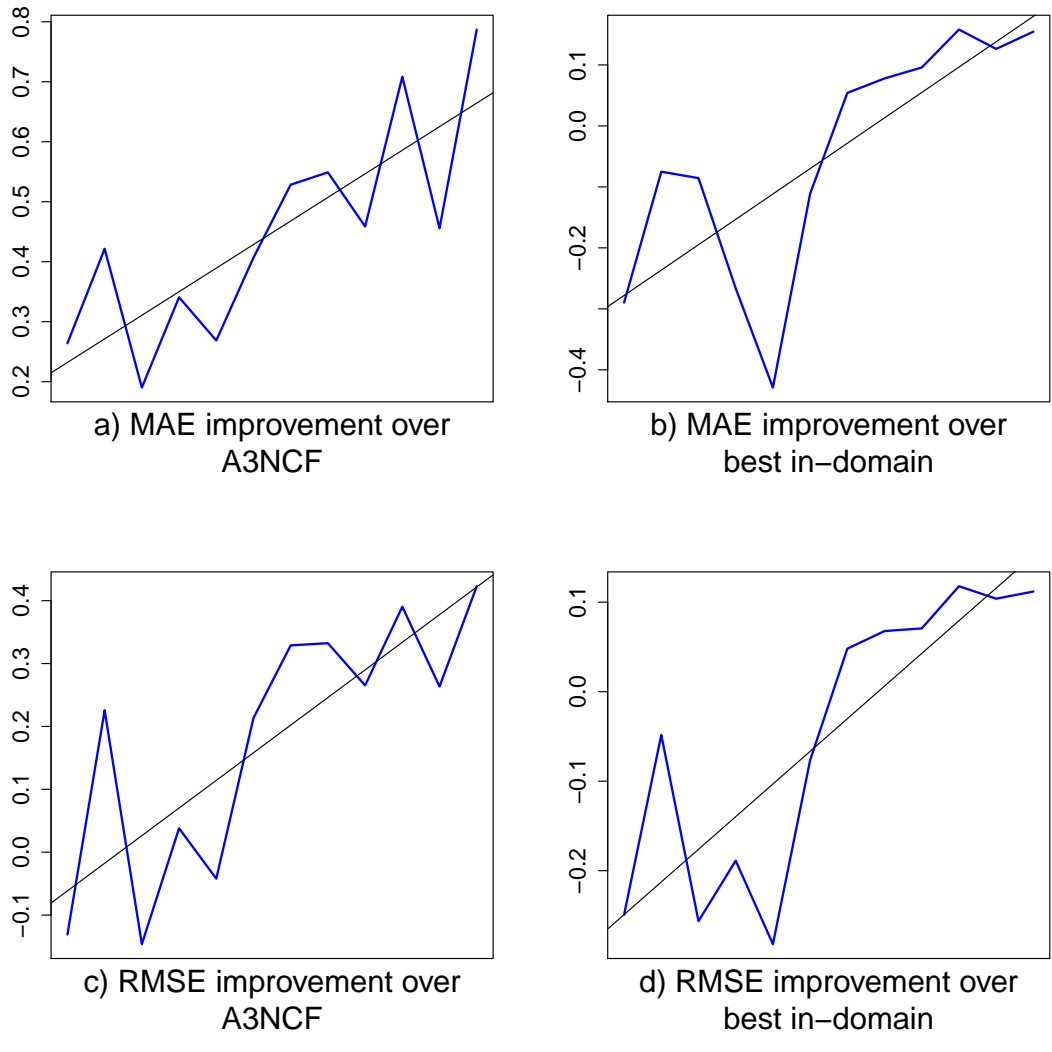

**Figure 2** **Relative improvement of the w2v+PAPE cross-domain classifier over the A³NCF and best in-domain (BID) classifiers.** Review domains are sorted by popularity on the *x* axis, with the least popular domains on the right. The relative improvement on each review domain is represented by the blue line. The line of best fit (in black) shows the overall tendency of our cross-domain classifier to achieve an increasing relative improvement over other methods as the popularity of the review domain and subsequently the amount of in-domain training data increases.

reinforces our cross-domain classifier as a suitable alternative to achieve the best results on the least popular domains, outperforming any other in-domain alternative.

## Effect of the size of the training data

Since cross-domain experiments benefit from larger training sets, we assess the effect of the size of the training data on the performance of the rating prediction system. We evaluate the performance of cross-domain rating prediction by using different percentages of the training data available, ranging from 1% to 100%, which we then compare with the best

**Table 7  MAE Results for cross-domain review rating prediction with different percentages of the training data.** BID, best in-domain. Categories are sorted in descending order by number of reviews, with the most popular review categories on top of the list.

| | MAE | | | | | | | |
|---|---|---|---|---|---|---|---|---|
| | 1 | 2 | 5 | 10 | 25 | 50 | 75 | 100 |
| books | 0.706 | 0.682 | 0.666 | 0.661 | 0.663 | 0.667 | 0.671 | 0.672 |
| | | | | BID = **0.521** | | | | |
| restaurants | 0.571 | 0.539 | 0.518 | 0.505 | 0.501 | 0.501 | 0.501 | 0.501 |
| | | | | BID = **0.466** | | | | |
| attractions | 0.615 | 0.604 | 0.596 | 0.594 | 0.593 | 0.595 | 0.596 | 0.596 |
| | | | | BID = **0.549** | | | | |
| clothing | 0.718 | 0.699 | 0.682 | 0.671 | 0.66 | 0.657 | 0.656 | 0.656 |
| | | | | BID = **0.518** | | | | |
| homeware | 0.869 | 0.848 | 0.818 | 0.800 | 0.787 | 0.779 | 0.777 | 0.776 |
| | | | | BID = **0.543** | | | | |
| hotels | 0.531 | 0.518 | 0.503 | 0.496 | 0.493 | 0.491 | 0.491 | 0.490 |
| | | | | BID = **0.441** | | | | |
| nightlife | 0.604 | 0.579 | **0.553** | **0.540** | **0.530** | **0.526** | **0.525** | **0.524** |
| | | | | BID = 0.554 | | | | |
| events | 0.601 | 0.575 | **0.550** | **0.537** | **0.527** | **0.523** | **0.522** | **0.521** |
| | | | | BID = 0.565 | | | | |
| casinos | 0.713 | **0.687** | **0.657** | **0.641** | **0.629** | **0.625** | **0.624** | **0.623** |
| | | | | BID = 0.689 | | | | |
| hair salons | 0.413 | **0.388** | **0.367** | **0.358** | **0.348** | **0.344** | **0.342** | **0.341** |
| | | | | BID = 0.405 | | | | |
| resorts | **0.685** | **0.658** | **0.632** | **0.619** | **0.606** | **0.604** | **0.603** | **0.602** |
| | | | | BID = 0.689 | | | | |
| dentists | 0.361 | 0.332 | **0.305** | **0.292** | **0.278** | **0.272** | **0.270** | **0.268** |
| | | | | BID = 0.317 | | | | |

in-domain rating prediction system. The subset of the training data is randomly sampled until the percentage is satisfied.

Tables 7 and 8 show the results of using different percentages of the training data for the rating prediction based on MAE and RMSE respectively. Results for each review category is compared with the best in-domain (BID) result, and the best results are highlighted in bold, i.e., either the best in-domain when this is the best, or the specific percentages when the cross-domain prediction system performs best.

As we observed before, the in-domain prediction system performs better for the top 6 review categories based on popularity, thanks to the availability of more training data. We are particularly interested in seeing how much data we need with the less popular categories for the cross-domain rating prediction systems to perform better than their in-domain counterparts. Looking at the 6 least popular review categories, we observe that using only 5% of the training data suffices in all cases to outperform the best in-domain system, with the percentage dropping up to 2% for casinos and hair salons and up to 1% for resorts. Note that out of the (up to) 58 million reviews available for cross-domain training, a subset

**Table 8  RMSE Results for cross-domain review rating prediction with different percentages of the training data.** BID, best in-domain. Categories are sorted in descending order by number of reviews, with the most popular review categories on top of the list.

| | RMSE | | | | | | | |
|---|---|---|---|---|---|---|---|---|
| | 1 | 2 | 5 | 10 | 25 | 50 | 75 | 100 |
| books | 1.311 | 1.281 | 1.262 | 1.258 | 1.264 | 1.270 | 1.276 | 1.278 |
| | | | | BID = **1.023** | | | | |
| restaurants | 0.976 | 0.925 | 0.895 | 0.876 | 0.869 | 0.869 | 0.869 | 0.868 |
| | | | | BID = **0.828** | | | | |
| attractions | 1.056 | 1.040 | 1.029 | 1.027 | 1.028 | 1.032 | 1.033 | 1.034 |
| | | | | BID = **0.823** | | | | |
| clothing | 1.259 | 1.236 | 1.217 | 1.206 | 1.194 | 1.191 | 1.190 | 1.190 |
| | | | | BID = **1.001** | | | | |
| homeware | 1.484 | 1.467 | 1.435 | 1.415 | 1.400 | 1.389 | 1.387 | 1.386 |
| | | | | BID = **1.081** | | | | |
| hotels | 0.907 | 0.889 | 0.865 | 0.854 | 0.848 | 0.845 | 0.844 | 0.844 |
| | | | | BID = **0.784** | | | | |
| nightlife | 1.022 | 0.987 | **0.951** | **0.933** | **0.919** | **0.914** | **0.913** | **0.912** |
| | | | | BID = 0.958 | | | | |
| events | 1.048 | 1.013 | **0.977** | **0.959** | **0.945** | **0.939** | **0.937** | **0.936** |
| | | | | BID = 1.004 | | | | |
| casinos | 1.146 | **1.115** | **1.078** | **1.058** | **1.045** | **1.040** | **1.039** | **1.039** |
| | | | | BID = 1.118 | | | | |
| hair salons | 1.015 | **0.969** | **0.929** | **0.910** | **0.890** | **0.882** | **0.878** | **0.876** |
| | | | | BID = 0.993 | | | | |
| resorts | **1.119** | **1.085** | **1.054** | **1.038** | **1.024** | **1.019** | **1.018** | **1.018** |
| | | | | BID = 1.136 | | | | |
| dentists | 1.058 | 1.004 | **0.953** | **0.925** | **0.896** | **0.884** | **0.878** | **0.873** |
| | | | | BID = 0.983 | | | | |

of 5% only represents (up to) 2.9 million reviews, which are easy to obtain through online review platforms.

We also observe that, while performance keeps improving as we increase the size of the training data, it generally shows a tendency to plateau after 25% of the reviews are used for training. This reflects that after about 14.5 million reviews the system has enough training data and can hardly improve its performance.

## CONCLUSIONS

In this work we have proposed a novel method that leverages aspect phrase embeddings for predicting the star rating of online reviews, which is applicable in in-domain and cross-domain settings. We have also shown that our cross-domain approach is effective for making predictions in review domains with a paucity of training data, where training data from other domains can be successfully exploited. Previous work on review rating prediction had been limited to popular reviews domains, such as restaurants or hotels. Our study broadens the findings of previous works by experimenting on 12 different datasets

pertaining to 12 different review domains of very different levels of popularity, and collects from different sources including Yelp, Amazon and TripAdvisor. Given that some of these review domains have very limited availability of labelled data for training, our aim has been to propose a cross-domain review rating prediction system that would perform well for those non-popular domains. Likewise, a cross-domain review rating prediction system can be used to predict ratings of reviews gathered from platforms where users do not assign ratings, such as Facebook or Twitter.

Our review rating prediction system leverages both POS taggers and sentiment lexicons to extract aspect phrases from reviews, i.e., phrases referring to different features of a business. To enable generalisation of aspect phrases to different domains, we make use of universal representations using word embeddings; we propose two different models for feature representations, (1) Aspect Phrase Embeddings (APE), which aggregates all aspect phrases of a review, and (2) Polarised Aspect Phrase Embeddings (PAPE), which considers positive and negative aspect phrases separately to create an embedding representation for each. We compare our results with those of the best-performing classifier by *Cheng et al. (2018)* and another baseline that uses word embedding representations of the entire review. We developed both in-domain and cross-domain review rating prediction systems following this methodology; this allows us to compare performance on in-domain and cross-domain experiments for different review domains.

Our experiments show that both of our methods leveraging phrase embeddings lead to improvements over the rest of the baseline methods, both in the in-domain and the cross-domain settings. Interestingly, a comparison of results for these two experiment settings shows that performance scores are higher for the in-domain classifier when we make predictions on the most popular domains, however the cross-domain classifier leads to substantial improvements for the least popular domains. Our results indicate that a classifier trained from in-domain data is more suitable for popular review domains, whereas unpopular review domains can be improved when out-of-domain data is used for training along with our aspect phrase embedding representation. Further looking into the amount of data needed for training a cross-domain rating prediction system, we observe that a small sample of about 5% of the data available (i.e., fewer than 3 million reviews) for the other domains is enough to outperform baseline methods.

Our work defines the state-of-the-art and the first approach to cross-domain review rating prediction, expanding the hitherto limited research focusing on in-domain settings for popular domains. Research in cross-domain review rating prediction is still in its infancy, and we hope that our work will encourage further research in the task. Future work on cross-domain review rating prediction could further focus on the detection of implicit aspect phrases for improving performance, as well as in capturing cultural differences in writing reviews, given that users from different countries may express themselves differently when writing positive or negative reviews. To further test additional methods to cross-domain rating prediction, our plans for future work include testing methods for transfer learning and domain adaptation.

### Funding
The authors received no funding for this work.

### Competing Interests
Arkaitz Zubiaga is an Academic Editor for PeerJ.

### Author Contributions
- Aiqi Jiang and Arkaitz Zubiaga conceived and designed the experiments, performed the experiments, analyzed the data, contributed reagents/materials/analysis tools, prepared figures and/or tables, performed the computation work, authored or reviewed drafts of the paper, approved the final draft.

### Data Availability
This Github repository is related to the code: https://github.com/SleepingAggie/star-rating-prediction.

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
