# Peer review of "Leveraging aspect phrase embeddings for cross-domain review rating prediction"

_PeerJ Computer Science, doi:10.7717/peerj-cs.225_

## Round 0.1 · original submission · Major Revisions

The reviewers have identified opportunities for improving the paper and, in particular, are concerned about the baselines used in the paper. More relevant baselines are needed.

Reviewer 1 ·

Basic reporting

The paper is well written, self-contained and easy to read.
Literature review is comprehensive.

Experimental design

The experimental design is appropriate.
However, I feel that a very relevant baseline is missing. Some background is required to make my point clear:

The two models in this paper create vectorial representations of reviews. These representations are built by concatenating a general word embedding-based representation (aggregating the embeddings of all words within the review) with aspect phrase embeddings (aggregating embeddings of words within phrases formed by opinions words + nouns or verbs). The opinion words are taken from a given opinion lexicon.

Proposed representations outperform the baseline that uses only a "general" word embedding representation. This is because representations coming from opinion words are being amplified in the final vector (they are included in the general embedding and the the aspect phrase embedding).

The paper would benefit a lot from further experiments including representations based purely on the opinion words. Perhaps simple representations that only use the embeddings of the opinion words or just the polarities of these words would work well.

It is hard to judge the contribution of this work without seeing these results.

Validity of the findings

Conclusions are well stated, but as discussed above, some additional experiments are required in order to judge the validity of the findings.

Reviewer 2 ·

Basic reporting

The paper presents an approach for rating prediction based on reviews leveraging on aspect phrase embeddings extracted from the reviews. This, in turn, enables the development of both in-domain and cross-domain review rating prediction systems. Particularly, the paper address an interesting issue within the rating prediction problem, which is the prediction for less popular, usually neglected domains. I think the problem of cross-domain in small domains is interesting and the experimental results suggest that there is an improvement in prediction for such domains.

Experimental design

The experimental results and the dataset used are correctly reported. The selection of the baselines need, however, more justification. The selection of the work of Fan and Khademi (2014) is hard to understand. It is a baseline using only text, which is the relevance of this particular work? And also why not to include other state-of-the-art works as baselines. The explanation of that other works made use of user data is not clear, which kind of data and why not to use it here?.

Validity of the findings

I think the conclusions can be more compelling about the contribution to the cross-domain prediction in small domains and include a discussion analyzing possible implications or even further improvements. For example, the results would be even better considering translating the knowledge from big domains (like hotels or restaurants) to small ones that are somehow related? What if domains can be hierarchically organized?

Additional comments

The paper have, however, some issues that can be improved. I believe the first three sections (Introduction, Background and Related Works) are a little misleading about clarifying the motivation and contribution of the paper. On the one hand, the authors emphasize the problem of the cross-domain in small domains, this problem is hardly discussed in the related works (which is the novelty of the paper in this regard? why not transfer learning or other approaches?). On the other hand, they also claim as a contribution to the extraction of aspect phrases mentioned in the text, but it is not clear the significance or previous works in adding phrases to aspect opinion mining. The three sections should be more focused toward highlighting the contribution(s) and the difference with other approaches. The second section may be even removed or merged with the other two.

---

## Round 0.2 · Minor Revisions

The submission looks good and is basically ready to be accepted, but there are some minor wording problems, etc., that need fixing. They are listed below and in the order in which they occur (first) in the document.

"as well as they are valuable" -> "and they are valuable"

"likert" -> "Likert"

"have the possibility for the contribution" -> "can also contribute"

"one is expected to use phrases" -> "reviewers are expected to use phrases"

"knowledge, the review rating prediction" -> "knowledge, review rating prediction"

"pursue the review rating prediction" -> "pursue review rating prediction"

"i.e. where phrases " -> "where phrases "

"system, which creates" -> "system that creates"

"found on the Internet, which do " -> "found on the Internet that do "

"consists in " -> "consists of " (multiple times)

"e.g. " -> "e.g., " (multiple times)

and the same for "i.e. "!

"the score of specific feature" -> "the score of specific features" (?)

"prediction, which are still limited " -> "prediction but are still limited" (?)

"did not study its effectiveness" -> "did not study their effectiveness"

"Subsequenty"

"In this work, we use " -> "We will use"

"except the cases" -> "except in the cases"

"we refer to aspect phrases" -> "we refer to as aspect phrases"

"the tuples conforming aspect phrases " -> "the tuples corresponding to aspect phrases " (?)

"that judges it " -> "that judges the object or action concerned"

"Representationn "

"two variants;" -> "two variants:"

"We generated the embedding vector" -> "We generate the embedding vector"

For the w2v baseline, how is a single vector created based on the entire text? By averaging all the embeddings?

"Where our objective is to build " -> "Because our objective is to build "

", which include " -> ":"

"on top of the list." -> "at the top of the list." (multiple times)

"where the combination of both" -> "whereas their combination with w2v"

"is also useful " -> "is useful "

"emphasises "

"popularity in the x axis" -> "popularity on the x axis" (twice)

Please include labels for the x axis in each of the sub plots in Figure 2.

"in-domain prediction system perform better " -> "the in-domain prediction system performs better "

"keeps improve " -> "keeps improving "

Make sure that all publications in the bibliography have page numbers and that the publisher is noted for conference proceedings!

---

## Round 0.3 · accepted · Accept

The submission looks fine now, but there is one new little problem in the second line of the introduction: "also can also contribute". Please fix that in the final version.